# Bioinformatics Insights on Viral Gene Expression Transactivation: From HIV-1 to SARS-CoV-2

**DOI:** 10.3390/ijms25063378

**Published:** 2024-03-16

**Authors:** Roberto Patarca, William A. Haseltine

**Affiliations:** 1ACCESS Health International, 384 West Lane, Ridgefield, CT 06877, USA; william.haseltine@accesh.org; 2Feinstein Institutes for Medical Research, 350 Community Dr, Manhasset, NY 11030, USA

**Keywords:** transactivation, HIV, SARS-CoV-2, TAR, Tat, nucleocapsid, RNA-dependent RNA polymerase, HEMIX, pan-coronaviral target, long COVID

## Abstract

Viruses provide vital insights into gene expression control. Viral transactivators, with other viral and cellular proteins, regulate expression of self, other viruses, and host genes with profound effects on infected cells, underlying inflammation, control of immune responses, and pathogenesis. The multifunctional Tat proteins of lentiviruses (HIV-1, HIV-2, and SIV) transactivate gene expression by recruiting host proteins and binding to transacting responsive regions (TARs) in viral and host RNAs. SARS-CoV-2 nucleocapsid participates in early viral transcription, recruits similar cellular proteins, and shares intracellular, surface, and extracellular distribution with Tat. SARS-CoV-2 nucleocapsid interacting with the replication–transcription complex might, therefore, transactivate viral and cellular RNAs in the transcription and reactivation of self and other viruses, acute and chronic pathogenesis, immune evasion, and viral evolution. Here, we show, by using primary and secondary structural comparisons, that the leaders of SARS-CoV-2 and other coronaviruses contain TAR-like sequences in stem-loops 2 and 3. The coronaviral nucleocapsid C-terminal domains harbor a region of similarity to TAR-binding regions of lentiviral Tat proteins, and coronaviral nonstructural protein 12 has a cysteine-rich metal binding, dimerization domain, as do lentiviral Tat proteins. Although SARS-CoV-1 nucleocapsid transactivated gene expression in a replicon-based study, further experimental evidence for coronaviral transactivation and its possible implications is warranted.

## 1. Introduction

Transcription of viruses requires a variety of viral and cellular proteins and is essential for viral genome amplification and reactivation. Viruses have a narrow window for effective replication and transmission. They have therefore developed strategies to boost their transcription, gene expression and immune evasion by affecting various cellular pathways, protein and nucleic acid interactions, and gene expression. Transactivation of gene expression, often encompassing intra- and transcellular effects, is one such strategy, also contributing to transmission, acute and long-term pathogenesis, and reactivation from latency. The human immunodeficiency virus (HIV)-1, for instance, encodes Tat, a transactivator of gene expression discovered four decades ago in the Haseltine laboratory. In cells infected with HIV-1, Tat stimulated, by at least two orders of magnitude, the synthesis of reporter genes placed under the control of the HIV-1 long-terminal repeat [1,2,3]. This activity was corroborated in an in vitro cell-free system [4,5]. Tat, thus, became the first non-prokaryotic transcription factor known to interact with RNA and emulate prokaryotic anti-termination factors, driving a positive feedback loop of gene expression.

HIV-1 Tat increases the processivity of the host’s RNA polymerase II [6,7] by recruiting the host’s super elongation complex (SEC) [8]. The SEC includes the heterodimer positive transcription elongation factor-b (P-TEFb) complex comprising Cyclin T1 and CDK9 [9,10,11,12,13,14], and the polymerase-associated factor 1 (PAF-1) complex [15,16] to the transacting responsive region (TAR). Located downstream of the initiation site for HIV-1 transcription (nucleotides +1 to +59), TAR adopts an A-form stem-loop structure with a bulge [4,17,18,19,20,21,22,23,24,25].

In the absence of Tat, short non-polyadenylated RNAs, terminating at TAR accumulate [26], are translocated to the cytoplasm, and translated into Tat [27,28]. Tat then enters the nucleus and nucleolus and binds P-TEFb, increasing its components’ efficiency, including triggering CDK9 autophosphorylation [9,10,29]. Levels of viral full-length polyadenylated long messenger (m)RNAs thus increase significantly [26,30,31,32] via several mechanisms [33].

The Tat-P-TEFb complex stabilizes RNA polymerase II, allowing it to overcome the premature termination of the transcription resulting from the assembly of two multi-subunit complexes, the negative elongation factor (NELF) and the DRB sensitivity-inducing factor (DSIF) [34,35], and RNA polymerase II meeting TAR [36,37]. The transition from abortive to productive elongation involves CDK9-mediated phosphorylation of the carboxyl-terminal domain of the largest RNA polymerase II subunit, RPB1. This occurs after Cyclin-T1’s binding to it, which is facilitated by liquid–liquid phase formation [7,10,38,39,40,41,42,43,44,45] and by phosphorylation of NELF and DSIF components [16,34,46,47,48].

Tat antagonizes other negative elongation factors. For instance, the positively charged arginine-rich TAR-recognition motif of HIV Tat shows strong similarity to the N-terminal part of the 7SK-binding motif of the hexamethylene bisacetamide (HMBA)-inducible (HEXIM) proteins from evolutionarily distant species [49,50]. Without involving Cyclin T1, Tat binds to a TAR-like sequence in the 7SK small nuclear RNA [51] and replaces HEXIM1 [50] in the 7SK ribonucleoprotein (RNP). This interaction disassembles the kinase-inactive 7SK/HEXIM/P-TEFb negative transcriptional regulatory small nuclear RNP, which includes up to half of the cellular P-TEFb [52]. The result is increased nuclear levels of active P-TEFb and viral transcription and replication. As another way to favor gene expression, when nucleosomes obstruct the path of the elongating RNA polymerase II, Tat-P-TEFb can modify histones. Tat-P-TEFB recruits histone acetyltransferases, like the CBP/p300 complex, to the viral promoter to activate nucleosome acetylation [37,47,53,54,55].

Beyond viral gene expression, HIV-1 Tat transactivates the expression of host cytokine genes [33], such as tumor necrosis factor (*TNF*)*-β* [56,57,58], interleukin (*IL*)*-6* [59,60], *IL-8* [61], the C-X-C receptor (*CXCR*)*4* chemokine [62], and the β-chemokine monocyte chemoattractant protein (*MCP*)*-1* [63]. Tat can also inhibit cytokine expression, as in the case of *IL-2* [64,65,66]. These effects are accomplished through TAR-like or non-TAR promoter regions that bind transcription regulators such as NF-κB and SP1 [14,67,68,69]. By binding to SP1, Tat can decrease the recruitment of SP1 to the promoter of the heat shock protein (HSP)70 binding protein (HSPBP)1 gene [70]. This decreases its expression and attenuates HSPBP1-mediated inhibition of viral replication [71]. Moreover, the cleavage and polyadenylation specificity factor (CPSF) interacts with the HIV-1 promoter to repress HIV-1gene expression. In turn, HV-1 Tat interacts with CPSF, attenuating said repression [72].

The effects of Tat on viral and host cytokine gene expression and their regulators underlie acute and chronic clinical manifestations. Tat can also be found as a surface-bound or an extracellular protein with an arginine–glycine–aspartic acid (RGD) integrin-binding motif encoded by a second exon, which does not contribute to the transcriptional activities of Tat but can contribute to neurological and vascular manifestations of HIV-1 infection [73,74,75,76]. For instance, HIV-1 Tat stimulates, in a TAR-independent manner, the expression of some endogenous retrovirus type K (HML-2) proviruses [77,78] or endogenous viruses linked to inflammation, neurodegeneration, and oncogenesis [79,80]. HIV-1 can also transactivate viruses such as the human neurotropic John Cunningham (JC) virus, a polyomavirus, via TAR-like sequences [81]. This reactivates JCV and increases the risk of developing progressive multifocal leukoencephalopathy, a neurodegenerative disorder.

Other than HIV-1, several viruses use the transactivation of gene expression. These include other lentiviruses, such as HIV-2 and simian immunodeficiency virus (SIV) [82,83], human T-lymphotropic virus types I and II [84,85], adenovirus [86], simian virus 40 [87], and human herpesviruses, such as human simplex virus [88] and Epstein–Barr virus (EBV) [89,90], among others. Viruses such as EBV can transactivate HIV-1 gene expression [89]. We, therefore, assessed, using a bioinformatics approach, if the severe acute respiratory syndrome-coronavirus SARS-CoV-2 and other coronaviruses might contain TAR-like sequences and, if so, if they also encode a Tat-like transactivator. We found both to be, in theory, the case. This is consistent with a study using a SARS-CoV-1 replicon [91]. Among all known proteins encoded by SARS-CoV-1, only the nucleocapsid significantly increased efficiency and accelerated, i.e., transactivated transcription when supplied in trans [91]. We discuss the potential implications of these findings, were they to be further validated experimentally, and other consistent evidence on replication, reactivation of self and other viruses, immune evasion, inflammation and cytokine storm, acute and chronic pathogenesis, and variant evolution of coronaviruses, which cause respiratory and systemic infections in humans and animals and are responsible for epidemics and pandemics.

## 2. Results

### 2.1. TAR-like Sequences Are Present in the SARS-CoV-2 RNA Leader

To determine if SARS-CoV-2 might also use gene expression transactivation, we first searched for a Tat-binding-like site in the SARS-CoV-2 leader and found the sequence AGAUCUG, which is identical to the HIV-1 and JCV TAR sequences to which HIV-1 Tat binds (AGAUCU, binding sequence underlined, with an additional guanosine residue), and spans the entire right stem half of stem-loop (SL) 2 up to the first nucleotide of SL3 (Figure 1). The secondary structure for the segment, including the AGAUCUG sequence of the SARS-CoV-2 leader sequence, is similar to that by which Tat recognizes TAR, i.e., with an indispensable first uridine in the bulge (AGA**U**CU) and G-C (A**G**AUCU) and A-U (AG**A**UCU) base pairs in the upper stem of the HIV-1 TAR stem–bulge–stem [21,22,23,25,92,93,94,95,96,97]. In JCV TAR, the Tat-binding sequence is at its loop, not its bulge.

Adjacent to the identical Tat-binding sequence in the SARS-CoV-2 leader, there are two other regions of similarity to the TARs of HIV-1 and JCV; the proximal one (light blue, Figure 1) spans most of the rest of SL2, and the distal one (fuchsia, Figure 1) spans most of the rest of SL3, including the transcription regulatory sequence (TRS). Although the SARS-CoV-2 leader lacks a Cyclin T1 binding sequence at the same position as HIV-1 TAR (loop), the complement of the Sarbecovirus SL1 loop and part of its second stem-half match the Cyclin T1 binding site sequence (7 of 10 nucleotides; Figure 1), which might provide a means for these coronaviral leaders to recruit Cyclin T1 if it can also bind the complement of its binding sequence. The JCV TAR also lacks a Cyclin T1 binding site at the same position as that of HIV-1 TAR but has a Cyclin T1 binding-like site in its bulge, which is also similar to the Sarbecovirus SL1 complement and is present in the HIV-1 TAR (Figure 1).

The similarities between the SARS-CoV-2 leader and the JCV and HIV-1 TARs are apparent not only at the primary structure but also at the secondary structure level. This is underscored in Figure 2, which shows the visualized secondary structures highlighting similarities among the stems of the hairpin (stem-loop) structures and the Tat-binding sites in the SARS-CoV-2 leader and the JCV and HIV-1 TARs.

### 2.2. The Leaders of All Genera of Coronaviruses Share the Same Regions of Similarities as SARS-CoV-2 with the TARs of HIV-1, HIV-2, SIV, and the Human 7SK Small Nuclear RNA

The regions of similarity between the HIV-1 and JCV TARs and the SARS-CoV-2 leader shown in Figure 1 are also present in the leaders of the other six β- and α-coronaviruses known to infect humans, in λ- and δ-coronaviruses, HIV-2 and SIV TARs and the human 7SK snRNA TAR (Figure 3).

Similarities shown in Figure 3 extend to β-coronaviruses infecting other species including, for example, bat (Accession numbers: MK211377, MK211378, KY770858, KY770859, KY417143, KY417146, KY417150) and civet (AY304486) for *Sarbecoviruses*; porcine (KY419113, DQ011855), rabbit (JN874562), bovine (NC_003405), rat (NC_026011, KM349742, NC_012936), and mouse (JQ173883, AF201929, AF208067, NC_001846) for *Embecoviruses*; and bat (KJ473821, NC_039207, NC_009019, DQ648794, EF065505, NC_009020, EF65510, EF065509) for *Merbecoviruses*. The regions of similarity are also well-conserved among SARS-CoV-2 strains (for instance, NC-045512, MT324062, MT123290.1, MT019530.1, MT020781.2, MT184908.1, MT152824.1, MT123291.2, MT163718.1, LC528233.1).

The HIV-2 leader has a 123-nucleotide-long TAR sequence relative to the 59-nucleotide-long TAR of the more pathogenic HIV-1 [103]. Depending on the model, the HIV-2 TAR can fold into a three-stem-loop structure or a long stem-loop. The Tat-binding site is located in a bulge region in both cases. There is heterogeneity among SIV TARs, as reflected in Figure 3. The TAR of the SIV CPZ strain is almost identical to that of HIV-1, while the TARs of the SIV MND and CPZ strains are closer in length and secondary structure to those of HIV-2. The Tat-binding sites also differ among the TARs of HIV-1 (AGAUCU; binding site underlined), HIV-2 (AGAUUG), and SIV (AGAUCU or GGGUUG depending on the strain). A similar heterogeneity in potential Tat-binding sites is seen among the leaders of the four genera of coronaviruses (Figure 3).

Leaders of coronaviruses other than those of SARS-CoV-2 and -1 do not have a region matching the complement of the Cyclin T1 binding site (with 7 or 6 nucleotides, respectively, out of 10 nucleotides; Figure 1). As shown in Figure 4, and in contrast to HIV-1 and JCV TARs (Figure 1), the similarity between the complement (antisense) of the HIV-2 TAR in the region corresponding to the Cyclin T1 binding site-like sequence (8 out of 10 nucleotides) extends to the sequence corresponding to the HIV-2 Tat-binding site (AGAUUG) within the SL1 region in the SARS-CoV-2 leader.

### 2.3. The C-Terminal Domain of the SARS-CoV-2 Nucleocapsid Protein Contains a Nucleotide-Binding Arginine-Rich and Adjacent Regions Similar to the TAR Binding Site and Adjacent Regions in HIV-1, HIV-2, and SIV Tat and the Human HEXIM Proteins That Bind to the 7SK snRNA TAR

The presence of TAR-like regions in coronaviral leaders led us to conduct an in silico search for a possible Tat protein homolog encoded by coronaviruses. As is the case with many RNA-binding proteins, the HIV-1 Tat protein contains an arginine-rich segment spanning amino acid residues 48–57 that bind TAR RNA through an arginine fork [104,105,106,107,108,109]. In general, arginine can interact with RNA nucleotide bases via highly specific hydrogen bonding and π-stacking, recognizing specific RNA stem-loops, internal loops, bulges, and purine quartets [22,94,108,110,111]. The basic region of Tat becomes partially or fully structured upon binding RNA and induces a conformational change in the RNA [107,108].

Using the BLAST algorithm, we compared the HIV-1 Tat protein and SARS-CoV-2 proteins, as well as amino acid sequences encoded by all six open reading frames in the sense and antisense SARS-CoV-2 genome. These analyses showed that the C-terminal domain of the nucleocapsid protein (N-CTD) of SARS-CoV-2 has a nucleic acid-binding, basic amino acid-rich region (amino acid residues 255 to 264) similar to the HIV-1 TAR-binding region (e = 0.02 for comparison between SARS-CoV-2 N and HIV-1 Tat; and 0.72 for comparisons between HIV-1 Tat and the translated sequence of sense open reading frame 2 of the SARS-CoV-2 genome). The similarity extends to adjacent regions of the SARS-CoV-2 N-CTD (Figure 5). Amino acids at both adjacent sides of the basic domain are important in increasing Tat–TAR binding affinity [94,112].

In Tat, the substitution of the basic amino acids lysine and arginine within the basic residue-rich binding region reduces binding affinity to TAR [112]. In the N-CTD of SARS-CoV-2, a 3D structural study revealed that some of the arginine and lysine residues in the basic amino acid-rich and adjacent region form a nucleic acid binding groove, which constitutes one of the most positively charged regions of the N protein [113]. The basic amino acids that form the groove can interact with the negatively charged single- and double-stranded RNA or DNA in vitro without nucleotide sequence specificity, with ssRNA molecules as short as 7 base pairs expected to be well-cradled by the groove [113,114,115,116]. For instance, a study showed that SARS-CoV-2 N bound a 17-mer single-stranded oligonucleotide (nucleotides 6-76: UGUCUCUAAACGAACU) spanning from the last base of the Tat-like binding site (underlined) to the end of the leader sequence [113].

Figure 5 shows that the five groove-forming basic amino acid positions within the basic amino acid-rich region are shared with the HIV-1 TAR (highlighted with red triangles), HEXIM proteins, and, with some exceptions, with the other β-, and α- λ-, and δ-coronaviruses. Some substitutions in the basic amino acid-rich region among coronaviruses are also seen in the HIV-1 Tat proteins of pandemic HIV-1 group M subtypes and other lentiviral Tat proteins [112]. The three basic amino acid positions outside the basic amino acid-rich region that were characterized to contribute to the nucleic acid-binding groove of SARS-CoV-2 are also present, albeit not as well-preserved among coronaviruses, lentiviral Tat proteins, and HEXIM proteins.

Other β- and α-coronaviruses infecting humans and λ- and δ-coronaviruses have similar N-CTD regions. The similarity extends to the Tat proteins of HIV-2 and SIV and to the host HEXIM proteins, which bind to the 7SK snRNA TAR-like structure (Figure 5).

No or a negligible percentage (<0.0004%) of amino acid substitutions in the TAR binding-like region of SARS-CoV-2 N-CTD were detected in a GISAID database [117,118,119] search of all SARS-CoV-2 lineages, including Alpha (B.1.1.7 + Q.*), Beta (B.1.351 + B.1.351.2 + B.1.351.3), Gamma (P.1 + P.1.*), Delta (B.1.617.2), Lambda (C.37 + C.37.1), Mu (B.1.621 + B.1.621.1), Omicron (B.1.529 + BA.*), BA.286 + BA.286.*, XBB.1.5 + XBB.1.5.*, XBB.1.16 + XBB.1.16.*, EG.5 + EG.5.*, BA.2.75 + BA.275.*, CH.1.1 + CH.1.1.*, XBB + XBB.*, XBB.1.9.1 + XBB.1.9.1.*, XBB.1.9.2 + XBB.1.9.2.*, XBB.2.3 + XBB.2.3.*, GH/490R (B.1.640 + B.1.640.*). Individual mutations were found in only poor-quality sequences.

The nucleocapsid protein is a substrate of cyclin-dependent kinase (CDK), glycogen synthase kinase, mitogen-activated protein kinase, and casein kinase II [120]. The binding of Cyclin T1 and CDK9 to the SARS-CoV-2 and -1 N proteins (amino acids 186 to 191) may recruit the P-TEFb heterodimer for the transactivation of gene expression.

Among known RNA-binding proteins, RGG/RG motif-containing proteins [110,120] are the second most common in the human genome [111,121,122]. However, lentiviral Tat proteins and the N-CTDs of SARS-CoV-2 and most other coronaviruses contain only one copy of an RG within or in the region adjacent to the TAR or TAR-like binding sites.

RNA binding of the SARS-CoV-2 genome is also required for viral capsid formation [123]. The N-CTD is also referred to as the dimerization domain, which is needed because of the instability of monomeric N-CTD [124]. However, the dimerization core required to form the viral capsid and distal to the nucleic acid-binding region is not shown in Figure 5 because it is not shared with lentiviral Tat. Moreover, it might not be needed for the potential transactivation activity of N-CTD that would be predicted to occur with N being part of the replication–transcription complex, including NSP12, during viral transcription and not assembly.

The arginine-rich domain in HIV-1 Tat not only functions as an RNA-binding domain (RBD) but also as a protein–transduction domain (PTD) and a nuclear localization signal (NLS) [125]. SARS-CoV-2 and -1 have a nuclear localization signal at the end of N, and localization may be regulated by phosphorylation of the serine–arginine-rich region of N [126,127]. The N coronaviral protein colocalizes with replicase components early during infection [128], suggesting its early intervention in RNA transcription [129,130,131].

### 2.4. The Interface Region of the NSP12 Protein of SARS-CoV-2 Contains a Cysteine-Rich Region That Is Similar to That Present in HIV-1, HIV-2, and SIV Tat Proteins

Another HIV-1 Tat-like feature revealed by the bioinformatics approach involves the highly conserved zinc-coordinating cysteine-rich region (7 cysteines out of 16 residues) that is essential for Tat function [132,133,134,135,136] via metal-linked dimers [137,138].

As shown in Figure 6, the interface region between the *Nidovirus* RNA-dependent RNA polymerase-associated nucleotidyl transferase (NiRAN) domain and the RNA-dependent RNA polymerase (RdRp) of the SARS-CoV-2 nonstructural (NSP)12 protein has the conserved cysteine-rich motif of HIV-1 Tat (e = 0.002 when comparing HIV-1 and SARS-CoV-2 NSP12). This domain is present in the Tat proteins of HIV-1, HIV-2, and SIV within a larger region of similarity. The positions of dissimilarity between the HIV-1/SIV and HIV-2 Tat proteins tend to coincide with those with dissimilarity between HIV-1/SIV Tat and the interface regions between NiRAN domains and the RdRps of coronaviruses. Although lentiviral Tat proteins have regions of similarities in two different coronaviral proteins, namely, the nucleocapsid protein and RNA-dependent polymerase protein, these two proteins, together with other nonstructural proteins, interact in the replication–transcription complex [139], where they may work in synchrony in the transactivation of gene expression.

### 2.5. The β-Arterivirus Porcine Reproductive and Respiratory Syndrome Virus Also Contains a Tat-Binding Site Identical to That in Human 7SK snRNA and Adjacent Similarities, and Its Nucleocapsid Has a Basic Amino Acid-Rich Region Similar to That in Tat

Arteriviruses have been grouped with coronaviruses under the order *Nidovirales*. The crystal structure of the SARS-CoV-1 nucleocapsid protein dimerization domain that partially overlaps with the nucleic acid-binding domain revealed an evolutionary linkage between corona- and arteriviruses [124]. We, therefore, analyzed the leader of the β-arterivirus porcine reproductive and respiratory syndrome virus (PRRSV) and found it to contain a potential Tat-binding site (CCAUUG) identical to that of human 7SK snRNA and closest to HIV-2 and SIV TARs with UU dinucleotides as binding sites (AGAUUG and GGGUUG, respectively) (Figure 7A). Regions of similarity, including the Tat-binding site, correspond to those shown in Figure 3. The coronaviruses with the closest similarities to HIV-1 Tat are the *Sarbecoviruses* SARS-CoV-2 and -1, with an identical AGAUCU Tat-binding site extending the similarity to the rest of SL2 and having, in SL1, a complement of the Cyclin T1-like binding site.

Figure 7B depicts the alignment of the PRRSV N with the Tat proteins of HIV-1, HIV-2, SIV, the human HEXIM1 and 2 proteins, and the coronaviral N-CTDs. The lentiviral Tat core region (highlighted in green before the red basic amino acid region in Figure 5) contains a lysine (K) residue at position 41 in HIV-1 and SIV-1 Tat, and 70 in HIV-2 Tat that is essential for transactivation, while the other residues in the core region are partially essential [133]. A lysine residue is similarly located in the N-CTD region of similarity of α-, β- and γ-coronaviruses analyzed and substituted for a negatively charged aspartic, acid, glutamic acid, or glutamine in human HEXIM1, δ-coronavirus, and the arterivirus PRRSV analyzed. The effects of such substitutions on binding affinities remain to be determined.

The zinc-coordinating cysteine-rich motif in lentiviral Tat proteins and the interface region of the coronaviral RNA-dependent RNA polymerase are absent in any of the known encoded proteins of PRRSV. These observations suggest that the transactivation of gene expression might extend to arteriviruses, and possibly to the entire *Nidovirales* order, illustrating how dissimilar RNA and DNA viruses develop similar strategies.

## 3. Discussion

The present study revealed the presence of TAR-like and Tat-like sequences in SARS-CoV-2, other coronaviruses, and an arterivirus infecting humans and animals. The TAR-like sequences are located in the leader regions and encompass stem-loop (SL) 2 and 3. The Tat-like sequences span the nucleic acid-binding and adjacent region of the CTDs of coronaviral nucleocapsid proteins and the N protein of an arterivirus, and a Zn-coordinating motif in the interface region between the NiRAN domain and the polymerase domain of coronaviral RNA-dependent polymerase (NSP12).

The findings in this paper, if validated experimentally, open up the possibility for coronaviruses and arteriviruses, both of the *Nidovirales* order, to join the ranks of other viruses that use the transactivation of gene expression as a means to favor their transcription and replication, and that of other viruses, while participating in immune evasion and viral-associated acute and chronic pathogenicity. However, determining static and temporally dynamic three-dimensional structural interactions among at least six viral proteins, RNA, and several host proteins in the replication–transcription complex to mechanistically understand a potential transactivation activity is not a simple task. It likely requires a combination of cutting-edge experimental approaches, as exemplified by recent studies on the complex protein-nucleic acid structural interactions that underlie the mitoribosome [140] and nuclear DNA damage repair machineries [141,142].

Despite the current lack of direct evidence, several experimental observations lend credence to the possibility of gene expression transactivation by coronaviruses. First, using a SARS-CoV-1 replicon expressing a luciferase reporter under the control of the transcription regulating sequence (TRS) for the gene encoding the membrane protein, a study showed that the nucleocapsid protein provided in trans significantly improved efficiency and accelerated, i.e., transactivated, transcription [91]. Interestingly, only the TRSs preceding the *M* and *N* genes of SARS-CoV-1 and -2 extend to include a potential Tat-binding-like sequence that, in the leader, is present at a similar position relative to the TRS-L, suggesting that transactivation might also occur directly at these TRS-B (body) sites. Second, the nucleocapsid and NSP12 proteins are part of the coronaviral replication–transcription complex, and although the domains in them with similarities to lentiviral Tat are not in the same protein, their presence in two proteins that are part of the same replication–transcription complex may allow both functions to be present as if they were in one protein. However, it is also possible that the presence of the Tat-like Zn-coordinating motif in NSP12 is simply irrelevant to a potential transactivation activity and only underlies other functions in NSP12, such as metal binding. Third, the nucleocapsid proteins of SARS-CoV-2 and -1 recruit similar transactivation-related host proteins, including Cyclin and CDK [120,143,144,145], as the lentiviral Tat proteins. Fourth, the RNA-binding regions of the N-CTDs of SARS-CoV-1 and other coronaviruses are positively selected [146], and the N protein of TEGV is involved in template switching during subgenomic RNA synthesis [130,147,148]. We will discuss, below, the potential implications of the transactivation of gene expression by coronaviruses if such activity is further validated experimentally.

Coronaviruses include pathogens of significant veterinary and medical importance. The emergence of a highly pathogenic human coronavirus in China in 2019 confirmed the long-held opinion that coronaviruses are important emerging and re-emerging pathogens [149]. The possibility of coronaviruses using the transactivation of gene expression as a tactic to hijack their host’s cell machinery to their advantage would be particularly pertinent to transcribing and replicating their complex single-stranded, positive-sense, non-segmented, polycistronic genomes up to 37 kilobases, the largest among RNA viruses [150].

The transactivation of gene expression may have an ancient origin, as do coronaviruses [151,152]. Coronaviruses currently infecting humans are believed to have emerged repeatedly from zoonotic sources for the past one thousand years [151,152,153]. Alpha coronaviruses are the oldest known coronaviruses infecting humans, and alpha and beta coronaviruses are estimated to have separated from the gamma and delta ones, for which wild birds are the main reservoir [154,155], 300 million years ago, when the evolutionary line for mammals was estimated to have diverged from that for birds [151,152].

The extended evolutionary periods have allowed coronaviruses infecting humans the opportunity to evolve and adapt to their host, rendering it particularly challenging to design preventive and therapeutic interventions that will be effective and durable. The transactivation of gene expression might, therefore, be part of an elaborate replication strategy, enabling coronaviruses to efficiently synthesize various proteins essential for the multiple stages of their life cycle and interactions with host cells. This might also render the transactivation of gene expression a therapeutic target.

### 3.1. Implications of Potential Transactivation to Transcription of SARS-CoV-2 and Other Coronaviruses

Transcription of SARS-CoV-2 and other coronaviruses takes place as a continuous process for open reading frame (ORF)1a/b in the proximal two-thirds of the genome and a discontinuous process unique among RNA viruses for the structural and accessory genes in the distal third of the genome [147]. The 5′ untranslated region (UTR) begins with a short 5′ leader sequence (nucleotides 1–69), added via discontinuous transcription to the 5′ end of all subgenomic RNAs encoding viral structural and accessory proteins. Although the 5′ UTR nucleotide sequences are somewhat divergent amongst coronaviruses, the secondary structure is highly conserved. SL2 is the most conserved structure in coronavirus 5′ UTRs [156]. Notably, SL2s from different coronaviruses can functionally replace each other, even across different genera [157]. We show that SL2 includes a Tat-like binding site and a proximal region of similarity with lentiviral and JCV TARs and human 7SK snRNA.

We also show that the nucleic acid-binding domain of N-CTD of SARS-CoV-2 and other coronaviruses is similar to that of lentiviral Tat and host HEXIM1 and 2, which are known to bind to the TAR-like structure of 7SK snRNA. The N proteins of human coronaviruses bind viral RNA and play pivotal roles in packaging and transcribing viral RNA [148,158]. The N gene is highly conserved and stable, with 90% amino acid homology and minimal mutations over time. Coronaviral N, a 45 kDa protein in SARS-CoV-2, is composed of two separate domains, an N- and a C-terminal domain, both capable of binding RNA in vitro in SARS-CoVs [114,115,159]. However, each domain uses different mechanisms to bind RNA, with N-NTD likely preferring unpaired single-stranded regions and N-CTD preferring double-stranded RNA. It has been suggested that optimal RNA binding requires contributions from both domains [160,161], creating a local high density of N [162,163,164,165].

The more distal region of similarity shown here between coronavirus leaders and lentiviral and JCV TARs comprises the coronaviral SL3, which contains the transcription regulatory sequence (TRS) essential for the synthesis of subgenomic messenger RNAs by discontinuous transcription [157]. Unwinding of SL3 by the N-NTD might be necessary for TRS function, as in mouse hepatitis virus, most closely related to SARS-CoV-1 and -2 [163,166]. Unwinding of SL3 by N-NTD might also contribute to N-mediated transactivation, while N-NTD binding to SL4a, SL7, and SL8 is involved in viral packaging [167].

The highly structured SARS-CoV-2 SL3 binds with high affinity to the human T-cell intracellular antigen (TIA)1 protein [168], a ubiquitous RNA-binding protein (RBP)20 that plays multifunctional regulatory roles in gene expression, including transcription, alternative splicing, and translation of messenger RNAs, as well as in cell stress and viral infections [169]. TIA1 protein could interact with conserved SL3 RNA elements within other beta coronavirus lineages [168]. How and if the NTD and CTD of the N protein interact with adjacent sequences in the coronaviral leader sequence and with other RNA-binding proteins, as well as the roles of these interactions on potential transactivation of gene expression and other functions, remains to be determined.

Our analysis also highlighted the presence of a shared zinc-coordinating cysteine-rich motif between lentiviral Tat and the interface region of the coronaviral RNA-dependent RNA polymerase protein. In lentiviral Tat, the cysteine-rich motif is required for dimerization, protein structure stabilization, and metal binding [109]. Beyond the RNA-dependent RNA polymerase (NSP12), the coronaviral replication–transcription complex includes processivity factors (NSP7-8), a helicase (NSP13), a single-strand binding protein (NSP9), a proofreading exonuclease (NSP14), other cofactors (e.g., NSP10), and capping enzymes (e.g., NSP16). In addition to many viral nonstructural proteins, the presence of cell nuclear proteins and the coronaviral nucleocapsid protein in the coronaviral replication–transcription complex increases virus amplification efficacy [170], to which the transactivation of gene expression might contribute based on the findings presented here. The complexity of the RTC, including N, is reminiscent of replisomes from DNA-based organisms and is potentially a consequence of the unusually large coronaviral genomes [170]. Extending the prokaryotic analogy, our comparisons of SARS-CoV-2 N-CTD against the GenBank database revealed its similarity (85.4%) to the amino-terminal portions of two bacterial site-specific DNA methyltransferases likely involved in epigenetic regulation of gene expression [171] (Figure 8).

Coronaviral N is the most copiously expressed protein during infection [172]. It is responsible for releasing nascent negative-strand RNA that promotes a template switch enabling the transcription of subgenomic RNAs [173]. Circularization or cyclization of the coronavirus genome, which places the 5′ with 3′ termini and their regulatory sequences [156,174] in proximity, has also been postulated as a necessary early requirement for the synthesis of subgenomic negative-sense strands [130] (Figure 8). In SARS-CoV-1 and -2, two complementary segments in the genomic termini have been identified [175]. The 5′ sequence is 36 nucleotides long. It begins at position 60 from the 5′ terminal m^7^G cap structure of SARS-CoV-2, which corresponds to the last uridine of the sequence in the SARS-CoV-2 leader identical to the HIV-1 Tat-binding sequence (AGAUCU). This 36-nucleotide segment includes the stem-loop structure SL3, which encompasses the TRS (ACGAAC) in the 5′ leader sequence and extends to the beginning of SL4. Another 36-nucleotide sequence complementary to the 5′ sequence is located at the 3′ terminus, immediately 5′ to the polyadenylation site (Figure 9).

In SARS-CoV-2, genome circularization would require the complete opening of SL3 and disruption of the triple helix junction in the 3′ UTR. In agreement with this observation, SL3 of related β-coronaviruses was suggested to be weakly folded or unfolded [176,177]. Genome circularization plays an essential role in the replication of several RNA viruses, including flaviviruses [175,178]. Competition within cells between intact and defective viral genomes may underlie the evolutionary selection of genome circularization by ensuring that only genomes bearing intact 5′ and 3′ UTRs engage with the replication machinery. The SARS-CoV-2 genome cyclization results in a complete opening of the 5′ SL3 where the TRS-L resides, raising the possibility that genome cyclization regulates SARS-CoV-2 discontinuous transcription, as was previously suggested for the mouse hepatitis virus [176]. In the setting of the linearized SL3, the N protein, by potentially binding through its CTD to the Tat-like binding sequence, may mediate the transactivation of subgenomic RNA expression.

The finding of potential circularization sequences only in the termini of SARS-CoV-2 and -1 RNA genomes raises the question of how other coronaviruses replicate without said sequences. In the bovine coronavirus, the N protein can act as a bridge to facilitate interaction between the 5’ and 3’ ends of its genome, leading to circularization of the genome [179]. A protein bridge composed of cap-binding protein, eIF4E, eIF4G, and poly (A) binding protein may also mediate circularization and subsequent initiation of replicase gene translation [180,181,182,183,184,185,186,187,188,189]. The presence of the circularization sequences in SARS-CoV-2 and -1 might help stabilize such protein bridges, potentially translate into increased replication competence, and contribute to the transactivation of gene expression of SARS-CoV-1 and SARS-CoV-2. Determining which viral and cellular factors contribute to the replication complexes and how the putative RNA-RNA interaction between genomic termini contributes to facilitating or stabilizing RNA–protein and protein–protein interactions in subgenomic RNA synthesis.

The discontinuous transcription process encounters a decision problem when reaching the TRS; transcription stops and switches to the leader TRS to produce shorter subgenomic mRNAs, or transcription continues through the TRS to generate longer subgenomic mRNAs and genomic RNAs. The greater abundance of shorter than longer subgenomic mRNAs suggests that transcription might switch upon reaching the 3′ end of the body TRS to produce shorter subgenomic RNAs [189]. However, the virus has to pass the body TRS in an appropriate proportion to produce enough longer subgenomic mRNAs and genomic RNA essential to the life cycle. Studies have investigated several *cis*-regulating elements and *trans*-regulating factors involved in this process [190].

The N protein might also participate in the discontinuous transcription of subgenomic mRNAs because the depletion of N from the replicon reduces the synthesis of subgenomic mRNA but not genomic RNA [130]. In the mouse hepatitis virus JHM strain, nucleocapsid phosphorylation and RNA helicase DDX1 recruitment enable the transition from discontinuous to continuous transcription [191]. Phosphorylation of viral N by host glycogen synthase kinase-3 (GSK-3) is required for template switching. GSK-3 inhibition selectively reduces the generation of genomic RNA and longer subgenomic mRNAs but not shorter subgenomic mRNAs.

N phosphorylation allows the recruitment of the RNA helicase DDX1 to the phosphorylated-N-containing complex, facilitating template readthrough and enabling longer subgenomic mRNA synthesis. DDX1 is a member of the DEAD-box protein family, the largest family of the superfamily 2 (SF2) helicases [192]. DDX1 knockdown or loss of helicase activity markedly reduces the levels of longer subgenomic mRNAs. DDX1 has been identified as a member of the cellular interactomes for the IBV N protein [193], suggesting that the interaction between DDX1 and N for regulating viral RNA synthesis could be a general phenomenon in coronaviruses.

The interaction between pS197-N and DDX1 could also function as a regulator to prevent the assembly of higher-order ribonucleoprotein complexes. Using a phospho-specific pS197-N Ab, a study [191] identified that the N protein is phosphorylated immediately after synthesis but is dephosphorylated in virions. Therefore, binding with pS197-N/DDX1 might impede packaging genomic RNA into mature virions. Similar dephosphorylation patterns for N proteins in assembled virions were observed in several other viruses [194,195,196]. Coronaviruses employ a unique strategy for the transition from discontinuous to continuous transcription to ensure balanced subgenomic mRNAs and full-length genomic RNA synthesis [148]. This transition would also influence N-CTD-mediated transactivation of gene expression.

As mentioned, coronavirus RNA synthesis is connected with the formation of double-membrane vesicles and convoluted membranes [128], thanks partly to the intervention of the N protein via liquid–liquid phase separation (LLPS). One of the RNA-binding proteins that host cells use to control viral infections is the RGG-containing motif cellular nucleic acid-binding protein (CNBP). CRBP binds SARS-CoV-2 RNA and competes with N to prevent the formation of liquid–liquid phase separation condensates for viral replication [197]. Phosphorylated CNBP also translocates to the nucleus and binds the *interferon-β* (*IFN-β*) enhancer together with IFN-regulatory factor 3 (IRF-3) to turn on the transcription of type I IFNs and antiviral responses. Conversely, the RNA-binding region in N-CTD inhibits IFN-β production, probably by shielding viral RNAs from recognition by cellular pattern recognition receptors (PPRs) [198].

The liquid–liquid phase separation that the SARS-CoV-2 N protein undergoes after binding to viral RNA recruits the key kinases of NF-κB signaling TAK1 and IKK complex, enhancing NF-κB activation [199]. The hCoV-OC43 N protein potentiates NF-κB activation by binding to one of its negative regulators, the microRNA (miR)-9 [200]. Infectious bronchitis virus and porcine epidemic diarrhea virus upregulate cFOS expression, and other AP-1 transcription factors, regulating coronaviral-induced apoptosis and favoring viral replication [201]. These effects can contribute to pathogenesis, such as inflammation and cytokine storm, by favoring the expression of cytokines, which could also be the potential transactivation activity of N.

The TAR-like features of the SARS-CoV-2 leader may mediate a competition with host 7SR snRNA, which would generate a higher proportion of the free form of the P-TEFb complex comprising Cyclin T1 and CDK available to bind to N and phosphorylate it, favoring its transport to the cytoplasm to participate in the transactivation of gene expression in compartmentalized settings.

Discontinuous transcription in coronaviruses involves transcription-regulating sequence (TRS)-dependent template switching, in which the leader TRS-L (ACGAAC core in β-coronaviruses) [199] interacts with homologous TRS-body (B) elements upstream of viral genes in the last third of the genome. An in vitro study using a SARS-CoV-1 replicon expressing a luciferase reporter under the control of a TRS-B derived from the region preceding the M gene showed that N provided in trans stimulated the replicon reporter activity, indicating that this protein may regulate coronavirus replication and transcription [91].

### 3.2. Implications of Potential Transactivation to Variant Evolution of SARS-CoV-2 and Other Coronaviruses

A systematic analysis [202] described yet another aspect of genome variation by β- and α-coronaviruses by documenting the presence of intragenomic rearrangements involving segments of the 5′ leader sequence in geographically and temporally diverse isolates of SARS-CoV-2 (Figure 10). The intragenomic rearrangements could modify the carboxyl-termini of the ORF8 (also in *Rhinolophus* bat *Sarbecovirus* β-coronaviruses) and ORF7b proteins; the serine–arginine-rich region of the nucleocapsid protein, generating the well-characterized R203K/G204R paired mutation; and two sites of the NiRAN domain of the RNA-dependent RNA polymerase (NSP12). Interestingly, the latter two rearrangements bring the Tat-like binding site and, therefore, the possibility of the transactivation of gene expression to a region of N preceding the one encoding the TAR binding-like segment of NSP12 preceding the Zn-coordinating domain shared with HIV-1 Tat.

Beyond SARS-CoV-2, similar rearrangements of 5′ UTR leader sequence segments, including the TRS-L, were found in all subgenera of β-coronaviruses except for *Hibecovirus* (possibly secondary to the availability of only three sequences in GenBank). These rearrangements were in the intergenic region between *ORFs 3* and *4a* and at the distal end of *ORF4b* of the *Merbecovirus* MERS-CoV; intergenic regions in the *Embecoviruses* hCoV-OC43 (between *S* and *Ns5*) and hCoV-HKU-1 (between *S* and *NS4*); and in the distal end that encodes the Y1 cytoplasmic tail domain of NSP3 of *Nobecoviruses* of African *Rousettus* and *Eidolon* bats. Intragenomic rearrangements were also found in α-coronaviruses in *NSP2* (*Luchacovirus* subgenus), nucleocapsid (*Nyctacovirus* subgenus), and *ORF5b* or *ORF4b* (*Decacovirus* subgenus). No rearrangements involving 5′ UTR sequences were detected for the β-coronavirus SARS-CoV-1; the other 12 subgenera of α-coronaviruses including hCoV-229E and hCoV-NL63 infecting humans; or δ (*Andecovirus*, *Buldecovirus*, and *Herdecovirus* subgenera) and γ coronaviruses (*Brangacovirus*, *Cegacovirus*, and *Igacovirus* subgenera).

The study, thus, highlighted an intragenomic source of variation involving duplication, inversion (in two α-coronavirus subgenera), and translocation of 5′ UTR sequences to the body of the genome with potential implications on gene expression and immune escape of α- and β-coronaviruses in humans and bats, causing mild-to-moderate or severe disease in endemic, epidemic, and pandemic settings. The intragenomic rearrangements involving 5′ UTR sequences which, in several cases, affect highly conserved genes with a low propensity for recombination, may underlie the generation of variants homotypic with those of concern or interest and with potentially differing pathogenic profiles. Intragenomic rearrangements are yet another example of the tremendous genomic flexibility of coronaviruses, which underlie changes in transmissibility, immune escape, and virulence documented during the SARS-CoV-2 pandemic [202]. The possibility that they may facilitate the transactivation of expression of specific subgenomic RNAs adds to their relevance. Increased levels of the nucleocapsid protein, as well as those of the overlapping reading frame OR9b and ORF6, have been associated with increased immune evasion of Alpha, Delta, and Omicron SARS-CoV-2 variants of concern [203,204].

### 3.3. Implications of Potential Intra- and Trans-Cellular Transactivation to Acute SARS-CoV-2 Pathology and Long COVID and Their Treatment

The N-CTD nucleic acid-binding domain may also transactivate other viruses containing TAR-like sequences, similar to the lentiviral Tat-binding domains to TAR and, as exemplified by the transactivation of JCV, by HIV-1 Tat leading to the development of progressive multifocal leukoencephalopathy. However, there would be subtleties in the affinity level of Tat- or Tat-like-binding to TAT- or TAR-like sequences. For instance, HIV-2 Tat binds TAR from HIV-1 with low affinity and only partially activates the HIV-1 LTR, while HIV-1 Tat binds TAR-1 and TAR-2 RNAs with similar affinity, fully transactivating HIV-1 and HIV-2 LTR [27,102,205,206,207].

HIV-1 Tat residues 30 to 55 interact with the SP1 transcription factor, mediate its phosphorylation, and increase HIV-1 LTR-driven gene expression [208,209]. N-CTD of coronaviruses may also bind SP1 based on the primary sequence similarities shown in Figure 5 between the HIV-1 Tat core and the basic amino acid-rich region with the nucleic acid-binding and adjacent regions of coronaviral N-CTD. The SARS-CoV-1 N-CTD is the main component of N to bind to the host translation elongation factor (EF)1α and inhibit host cell proliferation, which would also favor viral replication [198,210].

The effects of binding of coronaviral N-CTD to noncoding RNAs, as is the case with the binding of HIV-1 Tat to 7SK snRNA, may increase the levels of specific transcriptional factors; in the case of HIV-1, the P-TEFb complex composed of Cyclin T1 and CDK, which, in turn, activate gene expression of viral and cellular genes. Coronaviral N can bind to the Cyclin/CDK complex, which could render the need for a Cyclin T1 binding site in the coronaviral leader unnecessary. However, it is possible that the N-CTD binds to the Tat-binding-like site in coronaviral RNA and activates transcription by displacing host or viral proteins that may bind to SL2, which is highly conserved among coronaviruses, similar to the effects of HIV-1 Tat binding to 7SK snRNA.

Another mechanism that has been documented in SARS-CoV-2 and -1 to free active P-TEFb from the 7Sk snRNP complex involves the NSP7 protein, which, together with NSP8, assists the RNA-dependent RNA polymerase (NSP12) in the replication–transcription complex, with which N interacts, to catalyze the synthesis of viral RNA [211,212]. SARS-CoV-2 and -1 NSP7 interact with the 7SK small nuclear ribonucleoprotein (7SK snRNP) complex comprising La-related protein (LARP7), methyl–phosphate capping enzyme (MEPCE), and HEXIM1, which sequesters P-TEFb. The interaction between NSP7 and the 7SK snRNP releases active P-TEFb, which is critical for the replication of several viruses [213]. Based on the findings of the present study, coronaviral N-CTD could also bind to 7SK snRNA and free active P-TEFb, which could then bind to N. This possibility is further underscored by the finding of an HIV Tat-binding sequence identical to that in 7SK snRNA in the leader of the coronavirus-related arterivirus PRRSV and of a region in the PRRSV N similar to coronaviral N-CTD, including the TAR-like binding region (Figure 7). SARS-CoV-2 NSP7 shows no similarities to PRRSV proteins, lentiviral Tat, or human HEXIM1.

Beyond intracellular functions, the N proteins of SARS-CoV-2, hCoV-OC43, and mouse hepatitis virus are abundantly expressed on the surface of infected and neighboring cells, where the first two have been shown to inhibit leukocyte chemotaxis by binding to chemokines [214,215,216]. N can also sequester cytokines, possibly favoring viral transmission [214]. Like HIV-1 Tat, coronaviral N is exported from cells by a noncanonical process [217] and binds to the surfaces of other cells through electrostatic interactions with heparan sulfate proteoglycans.

Once in the extracellular compartment, HIV-1 Tat, through the same basic domain that, in the intracellular environment, binds to nucleic acids and is involved in translocation to the nucleus, binds to cell surface heparan sulfate proteoglycans, and is internalized by various cell types. This process requires the integrity of cell membrane lipid rafts and mainly occurs through caveolar endocytosis. This property is being used for the Tat-mediated delivery of fused heterologous proteins, nanoparticles, liposomes, phage and viral vectors, and plasmid DNA [218,219,220].

Extracellular Tat has been linked to chronic complications. Secreted Tat protein can be detected in cerebrospinal fluid, sera, and tissues of HIV-infected people, even in those without detectable viral load [221]. In addition to the HIV-1 surface glycoprotein gp160, extracellular Tat protein plays an important role in the development and progression of HIV-1-associated neurocognitive disorder (HAND), whose spectrum ranges from asymptomatic neurocognitive impairment, symptomatic mild neurocognitive disorder, to HIV-associated dementia [222].

HIV-1 Tat protein leads to the death of neuronal cells due to immune activation and rapid, as well as sustained production of cytokines, mainly TNFα, in macrophages and microglia [223]. There is a synergistic effect of HIV-1 Tat and TNF-*α* to promote neuronal death [224]. In our comparisons combining primary and secondary structures, we detected a similarity between the SARS-CoV-2 leader and the reverse sequence of the TNF-α leader, which may also result in a synergistic interaction between the SARS-CoV-2 and TNF-α leaders (Figure 11). In particular, the sequences of the loops in the SL1 and SL2 of the SARS-CV-2 leader are identical to those in the SL1-like and SL2-like loops and beyond (as highlighted in lighter colors in Figure 11) in the reverse TFα leader [225]. It might be possible that the SARS-CoV-2 leader also shares strategies to favor its gene expression with the TNF-α promoter if NSP1, which, by binding to SL1 protects SARS-CoV-2 RNA from degradation before its translation, also protects TNF-α RNA via its leader sequence.

The neurotoxicity of Tat protein is also linked with the presence of an R57 residue in the basic domain of Tat, and polymorphisms at this juncture can prominently increase the neurotoxic potential along with transactivation of the protein [221]. Tat secretion, therefore, represents an attractive target to attenuate HAND. The R57 residue is also present in the region of similarity with coronaviral N-CTD, except for the porcine delta coronavirus, where it is substituted for a glutamine (Q) residue.

SARS-CoV-2 replication might occur for several months after the initial infection, as evinced by the detection of subgenomic RNA, a marker of recent virus replication; the isolation of replication-competent SARS-CoV-2 from respiratory and non-respiratory tissues [226,227,228]; and the existence of viral reservoirs for SARS-CoV-2 [229,230]. More specifically, these studies have demonstrated the persistence of viral antigens, viral RNA, and whole virus in the brain, sinus, adrenal glands, kidneys, gut, lymph nodes, spleen, lungs, heart, and fungiform papillae in taste buds, which can underlie symptoms through direct viral cytopathic effects; regional inflammation; triggering an immune response causing an elevated and prolonged state of generalized inflammation; and prompting autoimmunity in pediatric or adult populations, independently from the severity of acute disease [231,232]. Extracellular coronaviral N could have similar effects as HIV-1 Tat in terms of toxicity to the nervous system or other organs, even though neither virus can infect neurons. Thus far, attention has focused on the extracellular SARS-CoV-2 S protein. Experimentally defining if SARS-CoV-2 includes the transactivation of gene expression will shed light on pathophysiological processes driven by its protein products in various organ systems.

In terms of treatment prospects, potential small molecule inhibitors of the Tat–TAR interaction have been identified over almost three decades; however, none has shown sufficient potency and selectivity [233,234,235,236]. HIV-1 nucleocapsid-fused chimeric proteins, namely nucleocapsid-HEXIM1-Tat and HEXIM1-Tat-nucleocapsid, have anti-HIV effects by inhibiting both HIV-1 transcription and packaging [237]. A similar approach could be used based on the SARS-CoV-2 nucleocapsid protein. However, the regulatory sequences shared among viruses and the multiple functions of host proteins involved pose a selectivity limitation as therapeutic targets.

The active form of P-TEFb, which drives transcription, elongation, and the transactivation of gene expression in HIV-1, and possibly also coronaviruses and arteriviruses, binds bromodomain-containing protein 4 (BRD4). Bromodomain and extra-terminal motif (BET) inhibitors induce P-TEFb release and are latency-reversal agents in HIV infection [212,238]. Latency-promoting agents could help manage the post-acute infectious syndromes seen with HIV-1, SARS-CoV-2 (long COVID), and other viruses and microbes.

Further insights into the workings of the nucleocapsid and the replication–transcription complex of coronaviruses and other nidoviruses are warranted to develop novel treatments for acute infection and to design latency-regulating strategies to deal with viral persistence and long-term consequences of infection. As discussed, three coronaviral proteins, namely, nucleocapsid through its nucleic acid binding-C-terminal domain interacting with Tar-like sequences in coronaviral leaders, NSP12 via the metal-binding, dimerization domain in the interface region preceding the RNA-dependent polymerase region, and NSP7 by allowing to free the P-TEFb complex from the 7SK ribonucleoprotein, would contribute to the potential transactivation activity of SARS-CoV-2, other coronaviruses, and the arterivirus PRRSV. All proteins are part of a larger coronaviral replication–transcription complex.

Further experimental evidence for the transactivation of gene expression in coronaviruses is warranted. Therefore, the implications of transactivation discussed here are merely hypothetical.

## 4. Materials and Methods

### 4.1. Detection of HIV-1 TAR-like Sequences in the SARS-CoV-2 Leader, and Expansion to Other Lentiviral, JCV, and Human 7SK snRNA TARs

We used manual inspection and the BLASTN program (Nucleotide BLAST: Search nucleotide databases using a nucleotide query (nih.gov); accessed from 1 October through 30 December 2023) [239] to determine the presence of a Tat-binding-like site and adjacent regions of similarity between HIV-1 TAR, the closely associated JCV TAR, and the leader of SARS-CoV-2 (β-coronavirus, genus *Sarbecovirus*). Upon finding regions of similarity, we expanded the search to include HIV-2 and SIV TARs, the human 7SK snRNA TAR, and the leaders of the six other β-coronaviruses (*Sarbecovirus* SARS-CoV-1, *Embecoviruses* hCoV-OC43 and -HKU1; and *Merbecovirus* MERS-CoV) and α-coronaviruses (*Setracovirus* hCoV-NL63, and *Duvinacovirus* hCoV-229E) known to infect humans, and of representatives of γ- and δ- coronaviruses, all associated with epidemics or pandemics.

We used the Rfam database (http://rfam.xfam.org/covid-19; accessed on 20 September 2023) with the curated Stockholm files containing untranslated region (UTR) sequences, alignments, and consensus RNA secondary structures of major genera of Coronoviridae; the representative RefSeq sequences for each genus obtained from the International Committee on Taxonomy of Viruses (ICTV) taxonomy Coronaviridae Study Group [240]; the reference sequences in the GenBank and NCBI Virus databases; and listings in publications involving phylogenetic analyses of alpha-, delta-, and gamma-coronaviruses from NCBI Taxonomy [241,242] to derive the leaders of coronaviruses of all genera for analysis. Variations in lentiviral TAR and Tat sequences were obtained from the literature.

Multiple alignments and sequence similarities derived from the manual, and BLASTN assessment were also evaluated using CLUSTAL Omega (https://www.ebi.ac.uk/services; accessed from 1 October to 30 December 2023) [243]. Primary structure alignments shown were selected based on shared secondary structural features that were determined based on alignment with lentiviral Tat, and as described next.

### 4.2. Secondary Structural Mapping of the Regions of Similarity between HIV TAR-like and Coronaviral Leader Sequences

Because lentiviral Tat binding involves both primary and secondary structural features of the TAR sequences to which they bind, RNA secondary structures of the HIV-1, HIV-2, SIV, and JCV TARs, the leaders of SARS-CoV-2, other coronaviruses, and the arterivirus PRRSV, and the human 7SK snRNA TAR, were derived from the literature and also visualized using forna, a force-directed graph layout (ViennaRNA Web services) [244,245,246]. Similar nucleotides were highlighted by designated color to the region of similarity to assess distribution in stem, loop, and bulge regions. Complementary (antisense) sequences were also analyzed, particularly those involving SARS-CoV-2/-1 SL1, because of the presence of the complement of a Cyclin T1 binding-like site in *Sarbecoviruses*.

### 4.3. Detection of HIV-1 Tat-like Sequences among SARS-CoV-2 Proteins

The HIV-1 Tat protein sequence was compared against SARS-CoV-2 proteins deposited in the GenBank^®^ database using the Basic Alignment Search Tool (BLAST)P^®^ (Protein BLAST: search protein databases using a protein query (https://nih.gov; Protein BLAST: search protein databases using a protein query (nih.gov); accessed from 1 October to 30 December 2023) [247] for SARS-CoV-2 and SARS-CoV-related viral proteins encoding similar stretches. All nonredundant translated CDS + PDB + SwissProt + PRF, excluding environmental samples from WGS projects, were searched, specifying severe acute respiratory syndrome coronavirus 2 as the organism. We also compared HIV-1 Tat against sequences in the six reading frames translated from the SARS-CoV-2 genome in the sense and antisense directions. The results of comparisons against known and uncharacterized open reading frames were compatible in terms of the significance of the similarity between HIV-1 Tat and the SARS-CoV-2 N-CTD domain or the SARS-CoV-2 interface region of the RNA-dependent RNA polymerase.

To assess the degree of conservation of the HIV-1 Tat-like N-CTD nucleic binding basic amino acid-rich region of SARS-CoV-2, we used the Global Initiative on Sharing All Influenza Data (GISAID) EpiFlu™ database of SARS-CoV-2 sequences (GISAID—Initiative; accessed from 1 November to 30 December 2023) [116,117,118], querying each amino acid position.

We extended the analyses to the HIV-2 and SIV Tat proteins, the human HEMIX1 and 2 proteins, the coronaviral N-CTDs, interface regions of RNA-dependent RNA polymerase of coronaviruses of the four genera, and the N protein of β-arterivirus porcine reproductive and respiratory syndrome virus. We used the same verification methods for multiple alignments of protein sequences as described above for nucleotide sequences.

## 5. Conclusions

The presence of TAR- and Tat-like sequences in SARS-CoV-2, other coronaviruses, and the arterivirus PRRSV, theoretically suggests that nidoviruses may use the transactivation of gene expression as another strategy to favor their transcription and replication using the host’s cell machinery. As summarized in Figure 12, three proteins would be hypothetically involved in transactivation, all part of the replication–transcription complex; namely, the nucleocapsid protein via its C-terminal domain with a nucleic acid-binding region similar to that in lentiviral Tats, NSP12 with a metal-binding, dimerization region in the interface region preceding the RNA-dependent RNA polymerase domain similar to that in lentiviral Tats, and NSP7 through its proven ability to bind to the 7SK ribonucleoprotein and liberate active P-TEFb.

Figure 13 depicts potential models for the binding of SARS-CoV-2 N to TAR-like leader sequences according to the binding of HIV-1 Tat to HIV-1 TAR or JCV TAR. For SARS-CoV-2, N-CTD might bind to the Tat-binding sequence, and Cyclin T1 to N or SL1. For HIV-1 TAR, the Tat binding and Cyclin-T1 binding sequences are almost contiguous, while for JCV TAR, they are further apart. For HIV-1 TAR, Tat binds to the bulge, while Cyclin T1 binds to the loop. For JCV TAR, Tat binds to the loop, while Cyclin T1 binds to the bulge. NSP12 does not contain a TAR binding domain and shares with HIV-1 Tat a metal-binding, dimerization motif. The NSP12 contribution to transactivation might be related to its RNA-dependent RNA polymerase activity, and the possible role of the shared metal binding dimerization motif in gene expression transactivation remains to be determined by further characterizing the replication–transcription complex structure and function.

An experimental study using a SARS-CoV-1 replicon showed transactivation by the SARS-CoV-1 nucleocapsid protein of a reporter gene under the control of the TRS-B of the gene encoding the membrane protein [91]. The mechanism was discussed as unknown, and the authors speculated on several scenarios, including the nucleocapsid protein’s ability to protect and increase the translation efficiency of viral RNA and stabilize replication/transcription complexes. However, the findings described here provide the possibility of direct transactivation of gene expression, even possibly directly proximal to the TRS-B preceding the gene encoding the membrane protein, which, similarly to the TRS-B preceding the nucleocapsid gene, contains a Tat-binding-like sequence. Transactivation of the expression of the membrane and nucleocapsid genes would be consistent with the observation that the nucleocapsid and membrane proteins are the most abundantly expressed by SARS-CoV-1 and -2.

Similarities to HIV-1 TAR and Tat were higher among *Sarbecoviruses*, *Embecoviruses* and *Merbecoviruses,* and included the presence of a region of similarity with the Cyclin T1 binding site in stem-loop 1 of the *Sarbecoviruses* SARS-CoV-2 and -1. Transactivation in nidoviruses might affect not only replication but also immune evasion, pathogenesis, transmission, and viral evolution. The approach used in this paper for sequence analysis combines primary and secondary structure features, including protein–nucleic acid binding interactions. Experimental validation of the bioinformatics findings described, and the potential implications discussed, is warranted.

## Figures and Tables

**Figure 1 ijms-25-03378-f001:**
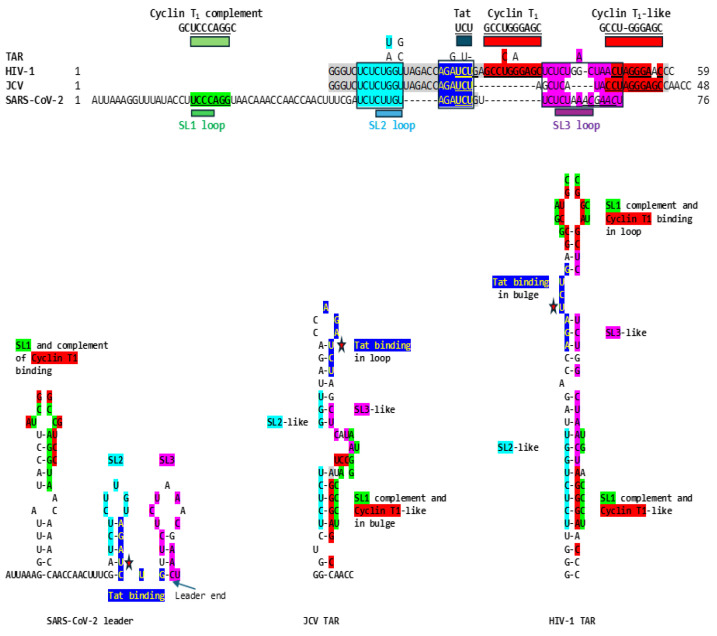
Similarities among HIV-1 TAR, JCV TAR, and the SARS-CoV-2 leader. The three regions of similarity are shown in light blue (SARS-CoV-2; NC_045512), dark blue (identical to the Tat-binding site in HIV-1 and JCV TARs also denoted with red stars in the bottom panel), and fuchsia in the primary and secondary structures. Differing nucleotides among HIV subtypes [98,99] are shown above the HIV-1 sequence. Cyclin T1 binding sites in HIV-1 and JCV TARs are shown in red. The complement of the HIV-1 Cyclin T1 binding site matching SL1 of SARS-CoV-2 is shown in green. SARS-CoV-2 TRS is underlined and in italics [100]. Matching nucleotides between HIV-1 and JCV TARs are highlighted in grey.

**Figure 2 ijms-25-03378-f002:**
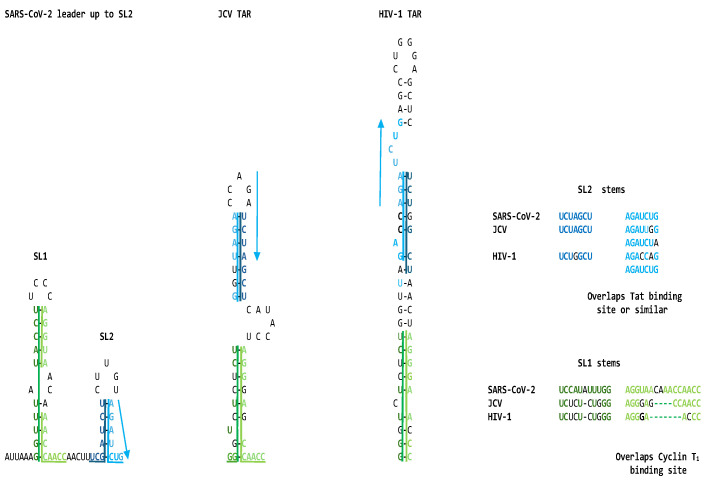
Similarities among the stems in the hairpin structures of the SARS-CoV-2 leader, JCV TAR, and HIV-1 TAR. Regions similar to the stems in SL1 and SL2 are shown using dark and light green and blue lines, respectively. As in Figure 1, green is used for SL1 and blue for SL2. Their primary sequences are aligned on the right. The light blue arrows depict the Tat-binding site in the three sequences, remaining putative for the SARS-CoV-2 leader.

**Figure 3 ijms-25-03378-f003:**
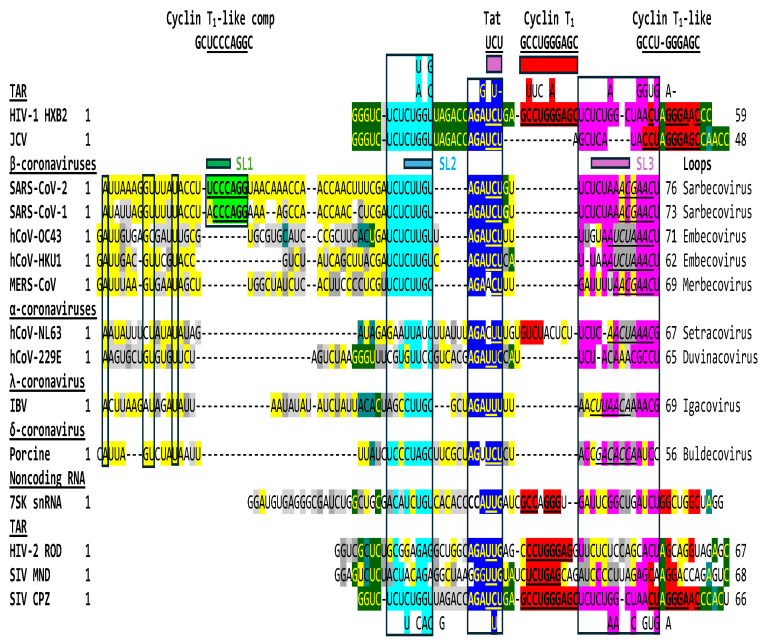
Conservation among the seven coronaviruses infecting humans and gamma- and delta coronaviruses of regions of similarity with lentiviral TARs and human 7SK snRNA. Nucleotide variation among HIV-1 TARs or SIV TARs are indicated at the top and bottom, respectively. Coronaviral TRSs [100] are in italics and underlined. The color scheme for the TAR-like regions of similarity is as in Figure 1. Similarities among sequences proximal to the shared TAR-like regions are highlighted in yellow and dark green. Accession numbers: SARS-CoV-2: NC_045512; SARS-CoV-1: NC_004718; hCoV-OC43: KJ958218; hCoV-HKU-1: MH940245; MERS-CoV: NC_019843; hCoV-NL63: MW202337.1; hCoV-229E: KU291448; IBV: NC_001451; Porcine deltacoronavirus: KR265862; human 7SK snRNA [50]; HIV-1, HIV-2 and SIV TARs [98,101,102].

**Figure 4 ijms-25-03378-f004:**
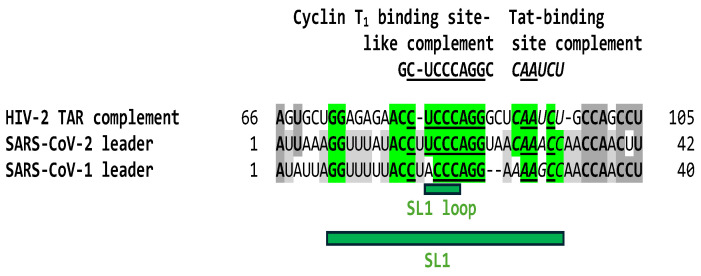
The HIV-2 TAR complement (3′ half of antisense corresponding to 5′ half of sense) shows regions of similarity (Cyclin T1 binding-like site and Tat-binding site; highlighted in green with other identical nucleotide positions highlighted in grey) with SL1 of SARS-CoV-2 and -1 (5′ half of sense), in contrast to HIV-1 and JCV TARs, which show regions of similarity to SL2 and SL3 (3′ half of sense). HIV-2 accession number: NC_001722.

**Figure 5 ijms-25-03378-f005:**
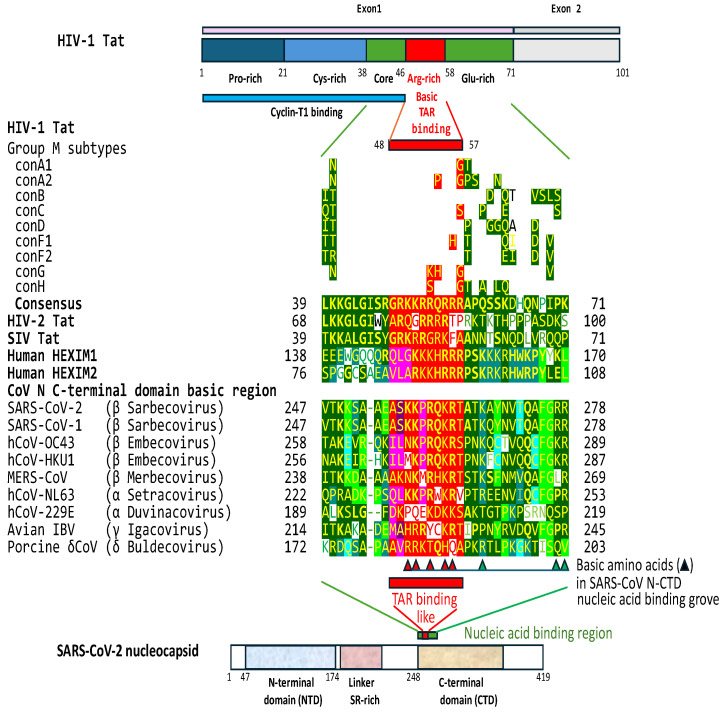
TAR binding-like region in the C-terminal domains of the nucleocapsid proteins (N-CTDs) of SARS-CoV-2 and other coronaviruses. The N-CTD region of similarity to lentiviral Tat and human HEXIM1 and 2 proteins is shown in red. Similarities in adjacent regions are shown in green. Differing amino acids among epidemic group M HIV-1 subtypes are shown above the HIV-1 Tat sequence [98]. Triangles indicate basic amino acids that form a positive charge groove for nucleic acid binding in SARS-CoV-2 N-CTD, with positions similar positions present in the TAR binding-like region highlighted in red or fuchsia and those in adjacent regions in dark or light green [113]. Accession: HIV-2: AAA76845.1; SIV: AEK79597.1; HEXIM1: NP_006451.1; HEXIM2: NP_653209.1; SARS-CoV-2 and -1: YP_009724397.2, YP_009825061.1; hCoV-OC43: AIX09803.1; hCoV-HKU1: AYN64565.1; MERS-CoV: YP_009047211.1; hCoV-NL63: UDL16983.1; hCoV-229E: AOG74787.1; Avian infectious bronchitis virus: NP_040838.1; Porcine deltacoronavirus: AML40885.1.

**Figure 6 ijms-25-03378-f006:**
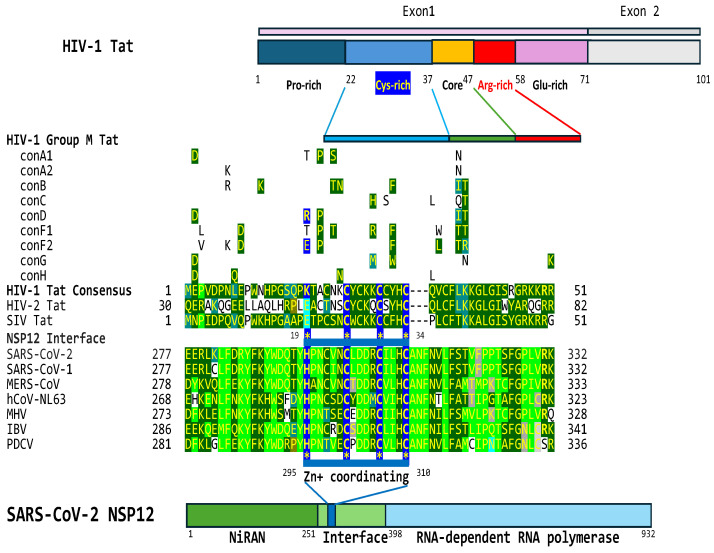
Similarity between zinc-coordinating domains of HIV-1, HIV-2, and SIV Tat (cysteine-rich region) proteins and the interface region of SARS-CoV-2 NSP12 and equivalent regions in other coronaviruses. The metal-binding motif consisting of cysteine and histidine residues in yellow with a dark blue background is conserved among coronaviruses. Cysteine and histidine residues involved in metal binding and that define the motif are highlighted with asterisks. ark green, blue-green, or teal is used for positions with similar amino acids among lentiviral proteins and to highlight shared similarities with coronaviral proteins, while light green is used for similar positions among coronaviral proteins. Differing amino acids among epidemic group M HIV-1 subtypes are shown above the HIV-1 Tat sequence [98].

**Figure 7 ijms-25-03378-f007:**
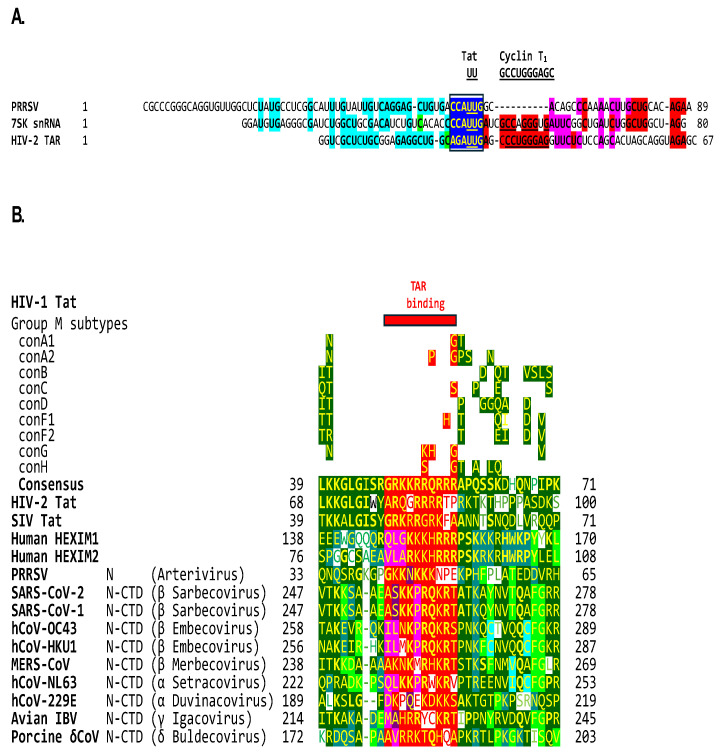
(**A**) Alignment of the PRRSV leader sequence (NC_001961) with human 7SK snRNA, and TARs of HIV-2 ROD strain and SIV MND strain. (**B**) Alignment of the PRRSV N protein (NP_047413.1) with lentiviral Tat proteins, human HEXIM1 and 2, and coronaviral N-CTDs. The color scheme for similar amino acid positions is as in Figure 5.

**Figure 8 ijms-25-03378-f008:**
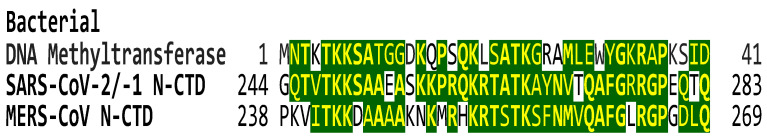
Alignment among a prokaryotic site-specific DNA-methyltransferase (*Candidatus Kaiserbacterium*) and the N-CTD nucleic acid-binding regions of SARS-CoV-2 and -1 and MERS-CoV. Identical (bold letters) and conservative substitutions are shown as yellow letters with a dark green background. Accession numbers for the bacterial sequence shown and for that of *Candidatus Nomurabacteria* are MCA9362859.1 and MCB9810153.1.

**Figure 9 ijms-25-03378-f009:**
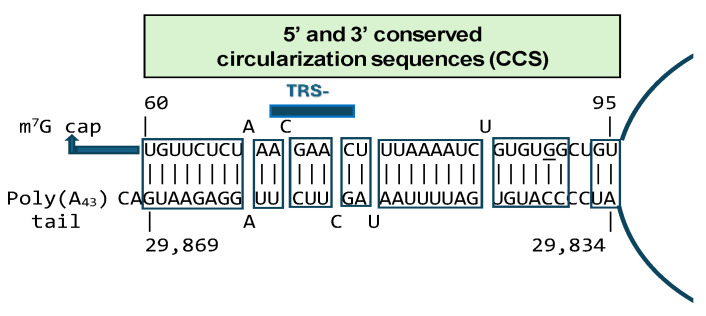
Conserved circularization sequences (CCSs) in 5′ and 3′ genomic termini of SARS-CoV-2 (Wuhan reference, NC_045512). The segment spanning between the last nucleotide of the potential Tat-binding site and the end of the SARS-CoV-2 leader encompasses half of the CCS in the SARS-CoV-2 leader. The underlined guanosine (G) is replaced by an adenosine (A) in SARS-CoV-1. The blue lines represent the beginning and end of the circularized viral genome.

**Figure 10 ijms-25-03378-f010:**
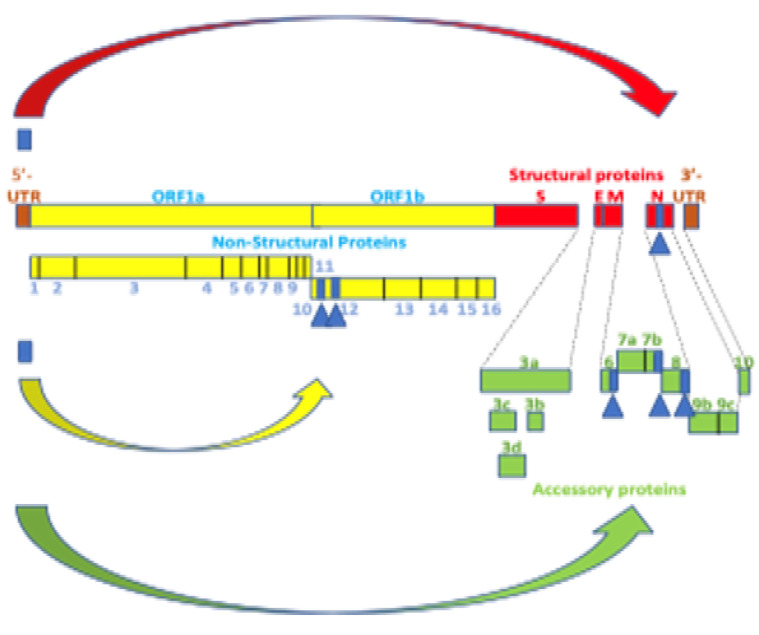
Intragenomic rearrangements in SARS-CoV-2 bring TAR-like sequences to regions preceding or within genes such as N and NSP12, preceding the lentiviral Tat-like sequences. Segments of the SARS-CoV-2 leader sequence encompassing the Tat-binding site and SL3, and, in some cases, the rest of SL2, are involved in intragenomic rearrangements in SARS-CoV-2, which would bring the Tat-binding site to regions usually preceding or within several viral genes. Genes encoding nonstructural proteins are shown in yellow, those encoding structural genes in red, and those encoding accessory genes in green. The leader segment is shown in blue, as are the positions of intragenomic rearrangements, which are highlighted with arrows in the colors assigned to nonstructural, structural, and accessory genes.

**Figure 11 ijms-25-03378-f011:**
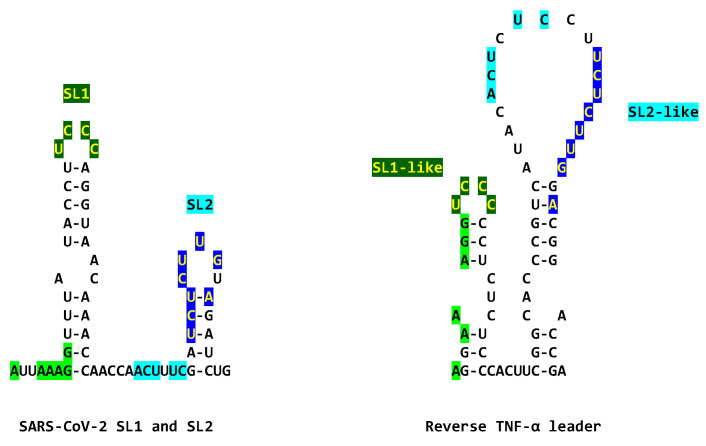
Comparison between SARS-CoV-2 SL1 and SL2 and the reverse of the TNF-α promoter. TNF-α sequence starts 3 nucleotides from the start codon to the transcription initiation site [225]. Regions of similarity involving SL1 sequences are shown in green, and those involving SL2 sequences in dark and light blue. The SL1 loop and similar region in the *TNF-α* leader are highlighted in yellow letters on a dark green background.

**Figure 12 ijms-25-03378-f012:**
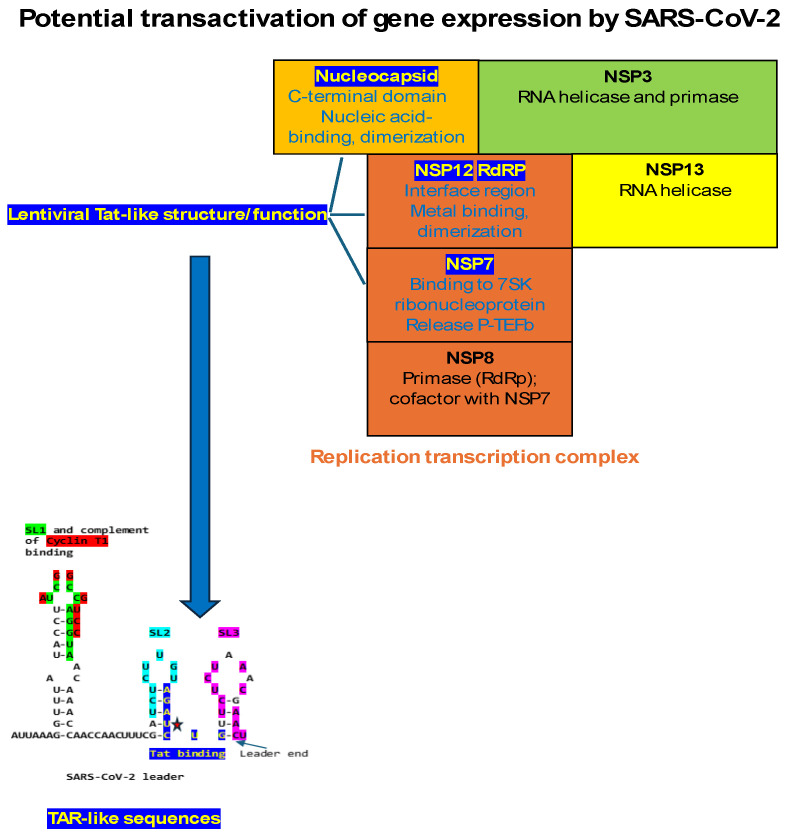
SARS-CoV-2 proteins with Tat-like structure and functions, and TAR-like sequences. Replication–transcription complex (orange) includes NSP12 (RNA-dependent RNA polymerase, RdRp), NSP7, and NSP8. The nucleocapsid protein (highlighted in light orange) interacts with NSP12 and NSP3 (highlighted in green). Beyond the nucleocapsid, NSP3 interacts with NSP12 and NSP13 (highlighted in yellow). The TAR-like sequences in the SARS-CoV-2 leader are indicated in light and dark blue in stem-loop (SL)2 and in fuchsia in SL3 (color scheme as in Figure 1). The complement of SL1 that matches the Cyclin T1 binding site in HIV-1 TAR is indicated in green. Star indicates the Tat binding-like site in the SARS-CoV-2 leader.

**Figure 13 ijms-25-03378-f013:**
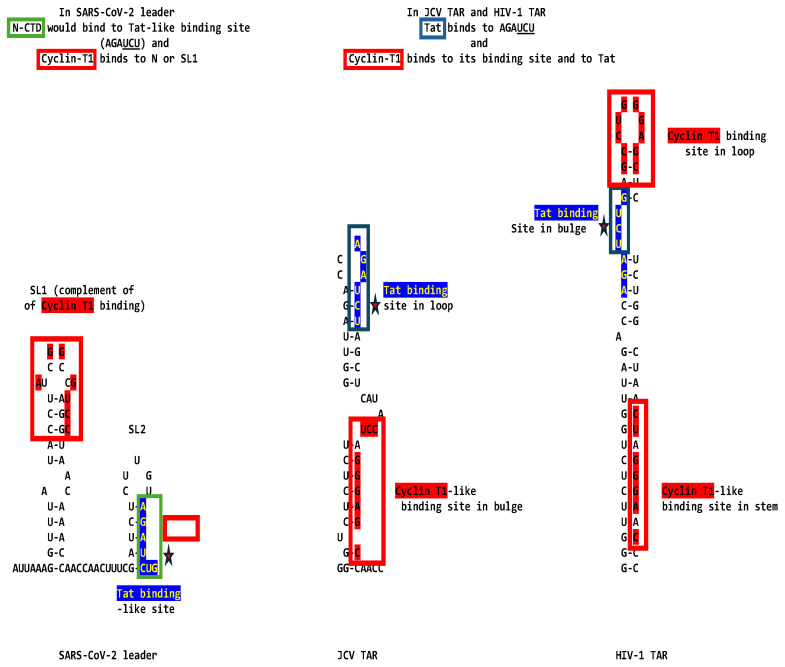
Potential models for the binding of SARS-CoV-2 N-CTD to TAR-like leader sequences, and HIV-1 Tat to HIV-1 TAR or JCV TAR. Cyclin-T1 binding and Cyclin-T1-like binding sites are highlighted in red. Tat binding or Tat binding-like sites are highlighted in blue, and with a red star.

## Data Availability

The data generated, and their sources are included in the article.

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
