# Peer review of "Bioinformatics Insights on Viral Gene Expression Transactivation: From HIV-1 to SARS-CoV-2"

_ijms, 2024, doi:10.3390/ijms25063378_

Round 1

Reviewer 1 Report

Comments and Suggestions for Authors

The manuscript by Patarca and Haseltine presents interesting findings showing that the SARS-CoV-2 genome (leader sequences) contains a TAR-like sequence, which may be involved in Tat binding. Such binding may transactivate viral and cellular RNAs in transcription and reactivation of self and other viruses via replication-transcription complex. This is interesting as Dr. Haseltine probably knows TAR and Tat interaction better than anyone else. It appears that more than bioinformatics, the experience of Dr. Haseltine played better in identifying TAR-like sequences in coronaviruses. I have only two comments. 

1. Identification of sequences is great. However, the structures of Tat-TAR complexes are available, which better define binding specificity. The authors should consider including a structural comparison of TAT-Tar complexes in different viruses.

2. The same applies to the Cys-rich sequence in nsp12. It is hard to imagine that nsp12 includes a TAR-binding-like structure in nsp12. The authors should consider including a structural comparison of the TAT-Tar complex from other viruses with the structure present in nsp12. 

Reviewer 2 Report

Comments and Suggestions for Authors

The authors have aligned HIV1 sequences with SARS-CoV-2 and other coronaviruses to demonstrate the presence of TAR-like sequences and similar TAR-binding regions of lentiviral Tat proteins in coronaviruses. 

The study seems to be preliminary and the conclusions just based on the alignments need to be carefully published and analyzed.

The authors are suggested to mention the significance and limitations of the study.  The authors have simply mentioned " If these implications prove valid, targeting the coronaviral transactivation might be apromising pan-coronaviral therapeutic target.". This concept needs to be elaborated to make the the study convincing.

Further studies are required to validate the conclusions.

Reviewer 3 Report

Comments and Suggestions for Authors

This article reported that the leaders of SARS-CoV-2 and other coronaviruses contain TAR-like sequences in stem-loops 2 and 3, the coronaviral nucleocapsid C-terminal domains (N-CTDs) harbor a region of similarity to TAR-binding regions of lentiviral Tat proteins, and the coronaviral nonstructural protein 12 has a cysteine-rich metal binding, dimerization domain similar to that in lentiviral Tat proteins using primary and secondary structural comparisons.

Several suggestions:

1.          lines 292-293, [The dimerization core required to form the viral capsid is distal to the nucleic acid-binding region shown in Figure 5.], No [dimerization region] is depicted in Figure 5.

2.       Please add a reference after [host proteins as the lentiviral Tat proteins] in lines 374-376.

3.       Though the findings in this article are really interesting, especially SARS-CoV-2 replicates in the cytosol while HIV replicates in the nucleus and then cytosol, the biological significance of these findings remains to be determined. It is better to have experimental results to prove any one of the speculations mentioned in the [discussion] section.

Round 2

Reviewer 2 Report

Comments and Suggestions for Authors

The authors have generously accepted the suggestion of putting the limitation of the study. However the findings are very preliminary and should be verified experimentally before getting published.

Author Response

Thank you for your reply. To address your comment, we underscored findings in what is now reference 91, which provide experimental evidence for the involvement of the nucleocapsid protein of SARS-CoV-1 in transactivating gene expression. We have added this study to the Abstract, Introduction, Discussion, and Conclusion sections while still underscoring that additional experimental evidence of transactivation and its implications is warranted.

In Abstract,

Although SARS-CoV-1 nucleocapsid transactivated gene expression in a replicon-based study, further experimental evidence for coronaviral transactivation and its possible implications is warranted.

In Introduction,

We, therefore, assessed using a bioinformatics approach if the severe acute respiratory syndrome-coronavirus (SARS-CoV)-2 and other coronaviruses might contain TAR-like sequences and, if so, if they also encode a Tat-like transactivator and we found both to be in theory the case. This is consistent with a study using a SARS-CoV-1 replicon that showed that among all known proteins encoded by SARS-CoV-1, only the nucleocapsid, when supplied in trans, significantly increased efficiency and accelerated, i.e., transactivated, transcription [91].

In Discussion,

Despite the current lack of direct evidence, several experimental observations lend credence to the possibility of gene expression transactivation by coronaviruses. First, using a SARS-CoV-1 replicon expressing a luciferase reporter under the control of the transcription regulating sequence (TRS) for the gene encoding the membrane protein, a study showed that the nucleocapsid protein provided in trans significantly improved efficiency and accelerated, i.e., transactivated, transcription [91]. Interestingly, only the TRSs preceding the M and N genes of SARS-CoV-1 and -2 extend to include a potential Tat-binding-like sequence that in the leader is present at a similar position relative to the TRS-L, suggesting that transactivation might also occur directly at these TRS-B (body) sites. Second, …

In Conclusion,

An experimental study using a SARS-CoV-1 replicon showed transactivation by the SARS-CoV-1 nucleocapsid protein of a reporter gene under the control of the TRS-B of the gene encoding the membrane protein [91]. The mechanism was discussed as unknown, and the authors speculated on several scenarios, including the nucleocapsid protein’s ability to protect and increase the translation efficiency of viral RNA and stabilize replication/transcription complexes. However, the findings described here provide the possibility of direct transactivation of gene expression, even possibly directly proximal to the TRS-B preceding the gene encoding the membrane protein, which, as does the TRS-B preceding the nucleocapsid gene, contains a Tat-binding-like sequence. Transactivation of expression of the membrane and nucleocapsid genes would be consistent with the observation that the nucleocapsid and membrane proteins are the most abundantly expressed by SARS-CoV-1 and -2.

Reviewer 3 Report

Comments and Suggestions for Authors

This revised manuscript has addressed most of the issues I raised previously though no experimental results are provided. Though the findings in this article are really interesting, the biological significance of these findings remains to be determined.

Author Response

(The authors gave the same response as above.)
